# Applications of Genome-Wide Screening and Systems Biology Approaches in Drug Repositioning

**DOI:** 10.3390/cancers12092694

**Published:** 2020-09-21

**Authors:** Elyas Mohammadi, Rui Benfeitas, Hasan Turkez, Jan Boren, Jens Nielsen, Mathias Uhlen, Adil Mardinoglu

**Affiliations:** 1Science for Life Laboratory, KTH–Royal Institute of Technology, SE-17121 Stockholm, Sweden; e.mohammadi@mail.um.ac.ir (E.M.); mathias.uhlen@scilifelab.se (M.U.); 2Department of Animal Science, Ferdowsi University of Mashhad, Mashhad 9177948974, Iran; 3National Bioinformatics Infrastructure Sweden (NBIS), Science for Life Laboratory, Department of Biochemistry and Biophysics, Stockholm University, SE-10691 Stockholm, Sweden; rui.benfeitas@nbis.se; 4Department of Medical Biology, Faculty of Medicine, Atatürk University, 25240 Erzurum, Turkey; hturkez@atauni.edu.tr; 5Department of Molecular and Clinical Medicine, University of Gothenburg, The Wallenberg Laboratory, Sahlgrenska University Hospital, SE-41345 Gothenburg, Sweden; Jan.Boren@wlab.gu.se; 6Department of Biology and Biological Engineering, Chalmers University of Technology, SE-41296 Gothenburg, Sweden; nielsenj@chalmers.se; 7BioInnovation Institute, DK-2200 Copenhagen N, Denmark; 8Centre for Host-Microbiome Interactions, Faculty of Dentistry, Oral & Craniofacial Sciences, King’s College London, London SE1 9RT, UK

**Keywords:** drug repositioning, genomic screens, machine learning, systems pharmacology, systems medicine

## Abstract

**Simple Summary:**

Drug repurposing is an accelerated route for drug development and a promising approach for finding medications for orphan and common diseases. Here, we compiled databases that comprise both computationally- or experimentally-derived data, and categorized them based on quiddity and origin of data, further focusing on those that present high throughput omic data or drug screens. These databases were then contextualized with genome-wide screening methods such as CRISPR/Cas9 and RNA interference, as well as state of art systems biology approaches that enable systematic characterizations of multi-omic data to find new indications for approved drugs or those that reached the latest phases of clinical trials.

**Abstract:**

Modern drug discovery through de novo drug discovery entails high financial costs, low success rates, and lengthy trial periods. Drug repositioning presents a suitable approach for overcoming these issues by re-evaluating biological targets and modes of action of approved drugs. Coupling high-throughput technologies with genome-wide essentiality screens, network analysis, genome-scale metabolic modeling, and machine learning techniques enables the proposal of new drug–target signatures and uncovers unanticipated modes of action for available drugs. Here, we discuss the current issues associated with drug repositioning in light of curated high-throughput multi-omic databases, genome-wide screening technologies, and their application in systems biology/medicine approaches.

## 1. Introduction

One of the first intentional and remarkable efforts for drug repositioning was carried out in 1987, when human immunodeficiency virus (HIV) became a full-blown pandemic with no known treatments. Prior to that time, azidothymidine (AZT) had been under development as an anticancer drug and had failed during clinical trials due to the absence of efficacy. Surprisingly, it took just over two years to move from the initial demonstration of AZT’s anti-HIV property to its FDA approval [1]. Empirical investigations of repositionable drugs have led to the rescue of the utility of many FDA-approved chemicals [2,3]. Empirical drug repositioning represents the process of finding an unknown target for a known drug in vitro. It has been proposed that empirical drug repositioning could be a starting point for the process of drug repurposing by the in vitro screening of known drugs or drug-like molecules, in order to identify and validate candidates for repositioning which result in the following advantages: The knowledge on a potential new disease setting will be increased; a serendipitous, or hypothesis-free, assessment of compounds will be achieved by testing multiple compounds with different modes of action; and data-driven choices will be produced for further investigations in more complex phenotypic or in vivo tests. As an example, an FDA-approved drug library of 640 compounds was screened and 10 of them, including tamoxifen and raloxifene, were proposed as compounds that are able to protect hair cell loss in response to known inner ear mechanosensory hair cell toxins, such as neomycin, gentamicin, kanamycin, and cisplatin, which cause hearing impairment and balance disorders [2]. In rational drug repurposing, the target is known from the beginning and subsequently, the goal is to find a previously FDA-approved compound which can interact with the target of interest [4]. In empirical drug repositioning, a large number of combinatorial compounds are prepared and tested. Rational drug repurposing reduces the number of candidates in the early stages of an investigation and identifies entities with a high probability of success [5]. Drug repurposing, also known as drug rescuing, repositioning, redirecting, tasking, recycling, or reprofiling, is a suitable approach for reusing existing drugs.

There are several reasons for the global need to repurpose currently approved drugs, instead of developing them de novo, including low success rates. At the beginning of 2020, there were 261,163 ongoing clinical trial studies (February 2020, www.clinicaltrials.gov) distributed all over the world (Figure 1a). In 2019, there were 318,901 ongoing clinical trials; while it is unclear how many of these clinical trials featured de novo drug design, only 48 novel drugs were approved for usage by the FDA (www.fda.gov). Taking this >95% failure rate into account, drug development is an extremely time- and resource-consuming process [3] insofar as the rate of success is almost 1 out of 10,000 (Figure 1b) [4]. 

On the other hand, whilst swift development for emerging diseases such as Covid-19 caused by SARS-Cov-2 is crucial, conventional procedures for drug development last more than 10 years [6,7]. FDA-approved drugs, or even those which have passed at least a phase I clinical trial (i.e., with proven human safety), could be considered as alternatives for rapid drug development. Considering that these drugs have passed at least a phase I clinical trial for one particular affliction, they could directly enter phase II or III of clinical trials for a second and unrelated indication. This path would take about two years, which is considerably shorter than the >14 years required for de novo drug development (Figure 1b), with an associated substantial cost reduction [3,8].

A single gene may be involved in two or more biological pathways or diseases and by targeting this gene, two unrelated diseases may be remedied. In addition, several mutations may disrupt the same biological pathway. In this case, all drugs which have similar effects on this pathway could be a candidate for repurposing studies [9]. For instance, trametinib is a mitogen-activated protein kinase kinase (MEK) pathway inhibitor, and is used for the treatment of leukemia and pancreatic cancer [9]. On the other hand, drug off-targeting could be another procedure employed for drug repositioning when the biological activity of a drug is different from its intended biological target. Despite being known as a common contributor to drug side effects, in some cases, off-target activity can be advantageous for therapeutic purposes. For example, the repurposing of the antimineralocorticoid and diuretic spironolactone, which is known to produce feminization and gynecomastia due to antiandrogen activity as side effects, could be used as an antiandrogen in the treatment of conditions like acne and hirsutism [10]. Additionally, sildenafil (Viagra®, Revatio®) [11] and minoxidil (Rogaine®) were originally developed for hypertension [12], but their off-target effects during trials were further explored in erectile dysfunction and promoted hair growth, respectively. Recent studies have aimed to make use of high-throughput datasets and computational approaches for the prediction of off-targets, in order to propose suitable repurposable drugs [13,14]. For instance, Rao et al. [13] developed the computational Off-Target Safety Assessment (OTSA) using more than 1 million compounds to identify potential off-target interactions that could be linked to predictable safety issues. In addition, Huang et al. [14] combined in vitro and in silico approaches to develop an integrative framework for the systematic identification of on-/off-target pathways and clarification of the underlying regulatory mechanisms.

Drug repositioning can be done based on prior knowledge or serendipity. This can be aided with the application of high-throughput screening, including CRISPR-Cas9 screening, RNAi Screening, and the application of systems medicine approaches for purposeful drug repositioning. One of the goals of genome-wide screening experiments is to generate and screen a population of genes to identify a particular phenotype or pathway and accordingly propose drug targets. The outcomes from screening studies could prepare the stage for systems medicine to integrate multi-omic data for accurate predictions of a drug’s mode of action, targets, and disease associations.

## 2. High-Throughput Resources for the Identification of Drug Targets and Repositioning of Drugs

Precision medicine aims to develop efficient therapeutic strategies with fewer side effects matching the unique disease signatures in specific cohorts, thus making use of precision pharmacology [15]. However, the accurate identification of suitable targets and repurposable drugs relies on databases (DBs) of chemical and biological compounds and high-throughput omic datasets. Here, we classified DBs indicated as suitable resources for the identification of drug targets and revealing the chemical–phenotype association according to the quiddity and origin of data, as well as whether they rely on computational- or wet lab-based approaches.

One may distinguish five groups of DBs [16] and their content (Figure 2) as follows: (i) Raw DBs, whose data are indirectly involved in drug repositioning and include literature curation, manual data uploading, data curation, integration, and clinical data; (ii) Target-based DBs containing drug targets and related information, including information on pathways, enzymes, side effects, genes, and protein targets; (iii) Specific DBs, which are those associated with specific software, disease data, and geographical information; (iv) Drug Design DBs, which typically include small molecules, 3D molecular structures, and molecular replacement information; and (v) Tool-based DBs, which embrace drug repositioning-related tools and include net-based tools (data mining through network theory) and simple tools (other approaches for data mining). These DBs rely on many different types of biological and chemical data and present different content availabilities and assays (Appendix A). Several DBs present particularly useful resources for data-driven analyses given their biological content, which often comprises high-throughput datasets (Table 1).

## 3. Computational- and Wet Lab-Based Assay DBs

Among the assay-based DBs, one may consider two general groups of databases based on the main approaches they use to obtain data, including computational- and wet lab-based assays. Computational-based databases are those which only benefit from in silico approaches, whether using primary assays such as data curation or advanced approaches, including network analysis. Wet lab-based datasets acquire data through wet lab assays.

### 3.1. Computational-Based Assay DBs

Various computational methods may be employed to acquire or collect biological data. For instance, text mining is a powerful technology for swiftly sifting key information through the vast quantities of biomedical literature. Text-mining techniques discover and extract new and unknown information from different resources, including databases, websites, books, reviews, and articles [39]. Manual data curation filters research results as they are generated, which means it unifies data for long-term sustainability and usage. In addition, this method monitors the reliability, reusability, and accessibility of obtained data [40]. Through integration, data from different sources may be processed and combined into a single, unified view [41,42].

Many datasets have followed one or a combination of these procedures, with or without experimental results, to design a comprehensive biological DB. For instance, GenomeRNAi (GRNAi) [43] is a manual extraction of phenotypes from RNAi screening in *Drosophila* and *Homo sapiens* literature, in addition to information about resources of RNAi reagents and their predicted quality. A dependency map (DMAP) [19] uses predictive modeling to collect and curate data from subdivisions of the DepMAP project, in order to propose gene–drug–disease associations. Gene Set Enrichment Analysis (GSEA)-MSigDB benefits from the gene set enrichment analysis method [42] and the Guide To Pharmacology Interactions (GPIs) [44], and Gene Set Database (GSDB) [45] benefit from data integration to present results.

In the case of methods used to retrieve drug–target associations, the Drug-Disease Network database (DNET) [46] uses differential co-expression analysis to obtain 1326 disease relationships among 108 diseases. Tissue- and cancer-specific biological networks (TCSBN) apply co-expression and integrated network analysis for 17 cancers and three tissues [33,47]. Additionally, Promiscuous (PROM) [48] organizes and curates data based on network concepts to predict new usages for existing drugs, considering the side effects. The Comparative Toxicogenomics Database (CTD) [49] provides manually curated information about chemical–gene/protein–disease associations, and the drug repurposing hub (DRHUB) [50] drug–gene interaction database (DGIdb) [51], DrugBank (DRUGB) [20], Small Molecule Pathway Database (SMPDB) [52], and Repurpose DB (RepDB) are comprehensive resources for gene targets, drugs and their categories, gene–drug interactions (DGIdb), and small molecule pathways (SMPDB). RepDB [53] is a compendium of drug targets, repurposed drugs, and their associated primary and secondary diseases. This database combines information on 253 drugs and 1125 diseases and using enrichment analysis, it determines key biological pathways, functional mechanisms, physicochemical features, and side effects associated with successfully repositioned drugs. These resources can help other researchers to design a better investigation for finding new targets for approved or even repositioned drugs.

Several repositories allow for manual and ununified dataset depositing, and include ArrayExpress [17] and Gene Expression Omnibus (GEO) [27], which contain functional genomics data from high-throughput functional genomics experiments. The Biological Magnetic Resonance Data Bank (BMRB) [18] contains experimental and derived data gathered from nuclear magnetic resonance (NMR) spectroscopic studies of biological molecules. BioModels [54] is a repository of mathematical models of biological and biomedical systems. In order to systematically analyze these repositories, some computational tools have aimed to integrate information for purposes that include proposing therapeutic targets. For instance, the Omics Discovery Index (OMDI) [32], as an open-source platform that enables access to and discovery and dissemination of omics data sets, may be used to integrate proteomics, genomics, metabolomics, and transcriptomics data sets from dozens of databases and repositories which have agreed on a common metadata structure framework and exchange format, and have contributed to OMDI. Similarly, the Metabolic Atlas [31] integrates open source genome-scale metabolic models. In addition, CTD utilizes various tools to discover the drug–target association through data provided by many datasets and repositories, including The Pharmacogenomics Knowledgebase (PhGKB) [55], DRUGB [20], geographic (GO) [56], and NCBI [57].

### 3.2. Wet Lab-Based Assay DBs

DBs containing information which is created through experimental approaches can provide the information required for many other projects and computational assay-based DBs. In this case, three different purposes might be considered. In the first group, projects may encompass large amounts of ununified raw data (i.e., Big Data), but this does not enable their direct application in drug repositioning. The second group includes projects which benefit from a specific approach and directly create relevant information about drugs, gene targets, protein targets, or any other data which clearly pave the way for drug repositioning. Finally, the third group is comprised of projects that aim to develop experimental methods in order to escalate the efficiency of experimental approaches.

The value of data provided by the first group is undeniable and large amounts of time and many efforts have been devoted to developing them. However, whilst most of the provided data are not directly aimed at drug repositioning, they may nevertheless be mined by advanced computational approaches. Many consortiums have achieved strikingly crucial advances in coordinating these efforts. For instance, the goal of the ENCODE consortium [23] is to build a comprehensive list of functional elements in the human genome, including elements that act at the protein and RNA levels, and regulatory elements that control cells and circumstances in which a gene is active. Systematically mapped regions of transcription, transcription factor associations, the chromatin structure, and histone modification in ENCODE have enabled researchers to assign biochemical functions for 80% of previously unknown protein-coding genomic regions. REMC [37], which is a similar project to ENCODE, aimed to identify DNA methylation, histone modifications, chromatin accessibility, and small RNA transcripts in primary human tissues. Another interesting example in this group is the LINCS L1000 [58,59] project, which developed its own method to quantify transcriptomic data from treated and untreated cancer cell lines. L1000 is a cost-effective high-throughput method that can be used to estimate genome-wide mRNA expression on a large scale based on the direct measurement of a reduced representation of the transcriptome (978 landmark genes), which captures most of the information contained in the entire transcriptome [58] and computationally infers the remaining transcriptome [27]. 

Other DBs enable user queries to identify drug targets and repositionable drugs. For instance, the Achilles (ACHI) project, which is one of the subdivisions of DMAP and provides its essential data, systematically identifies and catalogs gene essentiality in genomically characterized cancer cell lines through genome-scale RNAi and CRISPR-Cas9 genetic perturbation reagents, in order to identify those genes that affect cell survival [21]. We discuss RNAi and CRISPR-Cas9 screens below. Identifying and targeting cancer dependencies with small molecules is the aim of the Cancer Therapeutics Response Portal (CTR) [60], which has accelerated the discovery of patient-matched cancer therapeutics. CTR has generated a set of 481 small-molecule probes and drugs that selectively target distinct nodes in cell circuitry and that collectively modulate a broad array of cell processes. In addition, the sensitivity of 860 deeply characterized cancer-cell lines toward Informer set compounds is quantitatively measured and connected to cancer features, including mutations, gene expression, copy-number variation, and lineages, and relevant results are freely available on the CTR database [61]. The Cancer Cell Line Encyclopedia (CCLE) provides public access to genomic data, analysis, and visualization for cancer cell lines [19]. It is one of the many large panels of comprehensively characterized human cancer models and helps to study genetic variants, candidate targets, and small-molecule and biological therapeutics. In addition, it may be used to identify new marker-driven cancer dependencies by examining genetic, RNA splicing, DNA methylation, histone H3 modification, microRNA expression, and reverse-phase protein array data for 1457 cell lines [19]. The Genomics of Drug Sensitivity in Cancer (GDSC) DB characterized many human cancer cell lines and screened them with hundreds of compounds, and accordingly, made drug response data and genomic markers of sensitivity available [26].

Among the projects that aim to escalate the experimental efficiency is the National Center for Advancing Translational Sciences (NCATS) consortium at NIH. The aim of this consortium is to develop, demonstrate, and disseminate tools and solutions to be used by all translational investigators. NCATS arms researchers with translational science information by making new methods, data, and information publicly available. For instance, the Assay Guidance Manual (AGM), as a procedure for data transparency and release in NCATS, resulted in the successful development of robust, early-stage drug discovery assays [62]. In addition, in the case of the availability of data, NCATS serves the pharmacological industry through PubChem [63] (small organic molecules and their activities against biological assays), the NCATS pharmaceutical collection [64] of active pharmaceutical ingredients and small molecules, and small molecule probes [65].

Another example of efforts aiming to escalate the experimental efficiency is the design and implementation of high-throughput G-protein-coupled receptor (GPCR) assays in GPCR Assay Bank that allow the cost-effective screening of large compound libraries, in order to identify novel drug candidates [66]. GPCRs are some of the most successful therapeutic target assays for a broad spectrum of diseases and mediate many important physiological functions. The GPCR Assay Bank is the largest readily accessible collection of approved drugs and has several subdivisions (https://web.duke.edu/gpcr-assay/my_screening.html), including Johns Hopkins Clinical Compound Library (JHCCL) (1514 compounds), KinaseGold Library (3893 compounds), PresTwick Library (1120 compounds), Sigma Kinase Library (97 compounds), Steroid Drug Library (1659 compounds), Tripos Library (50K compounds), TimTec NPL (Natural Product Library, 720 compounds), ActiProbe-5K (5K compounds), and Chemdiv Library (https://www.chemdiv.com/) (299,408 compounds).

## 4. Application of Genome-Wide Screening in Drug Repositioning

Functional genetic screening connects genes or genetic elements to phenotypes-of-interest. In this case, several methods have been gradually developed [67]. RNA interference (RNAi) oligonucleotides for loss-of-function studies [68] and cDNA overexpression libraries for gain-of-function studies [69] were performed prior to the clustered regularly interspaced short palindromic repeats (CRISPR)-associated Cas9 endonuclease system [67] and systems biology approaches [70]. All of these methods, especially CRISPR-Cas9, revealed a wealth of mechanistic insights, from drug resistance in cancer to neuronal toxicity in amyotrophic lateral sclerosis, and also made prerequisite information for drug repositioning by introducing drug targets available.

### 4.1. Drug Target Identification through RNA Interference Screening

RNA interference (RNAi) is a post-transcriptional gene regulating system which benefits from small noncoding RNAs and is utilized for the sequence-specific knockdown of a gene’s function [4,71]. Small interfering RNAs (siRNAs), individually or as a pool of synthetic RNAs, are designed to specifically target unique messenger RNAs (mRNA) for a period of time or permanently [72]. High-throughput RNAi screening leads to the discovery of gene functions and its applications have been proven in many fields, such as infection, cancer, obesity, and aging.

RNAi can be introduced to the cell in various ways (Figure 3a), but all of them will end up with the siRNA duplex(es) entering the RNA-induced Silencing Complex (RISC) pathway. RISC can unwind and cleave the RNAi and realize the guide strand to pair with the target mRNA via perfect sequence complementarity. Afterwards, the RISC which is activated and specified by guide strand cleaves the target mRNA and accordingly, the protein production is abolished [73].

In Figure 3b, we provide a general workflow of siRNA screening towards drug repositioning. After transfection of the cell culture system with an siRNA library, a particular disease model (e.g., specific cancer) is induced in the cultured cells. Pre-determined endpoint assays are conducted at desired time points to yield hits (the genes which have an influence on disease) and lists of gene hits are compiled. Subsequently, these genes are validated in various conditions, such as different cell types, viruses, inductions, and different end-point assays. After the detection of a target gene, validation can be conducted by single siRNAs instead of siRNA pools. Eventually, these genes can be potentially targeted as a therapeutic disease intervention strategy [72].

Among the successful experiments involving genome-wide RNAi screening is the application of a library of 74,905 retroviral shRNAs targeting 32,293 unique human transcripts to determine synthetic lethal interactions with the Kirsten rat sarcoma viral oncogene homolog (KRAS) oncogene, which is the isoform most commonly mutated in human cancers [74]. More recently, Takai et al. carried out a genome-wide RNAi through the GeneNet h50K shRNA library, which is composed of ~200,000 lentiviral shRNAs and more than 47,000 transcript targets, in order to identify genes with a synthetic lethal interaction with epithelial cellular adhesion molecule (EpCAM) as a potential therapeutic target for the EpCAM + AFP + Hepatocellular carcinoma subtype [75].

RNAi high-throughput screening, in spite of allowing for the swift discovery of the molecular basis of many diseases, and the identification of potential pathways for developing safe and effective treatments [72], presents issues associated with off-targeting [71].

### 4.2. Drug Target Identification through Genome-Wide CRISPR-Cas9 Screening

CRISPR-Cas9 genome-wide essentiality screens present suitable technologies for improving target prediction while minimizing off-targeting, thus improving on RNAi screens. Repurposing CRISPR as an RNA-guided platform for the sequence-specific control of gene expression was developed in 2013 [76]. Since then, many efforts have been made to conduct gene screening through this approach [9,77,78]. CRISPR-Cas9-based technology allows the discovery of efficient and precise new or repurposed drug candidates for genomic or proteomic targets. Systematic screenings of genes associated with a disorder can be conducted by combining the Cas9 endonuclease and a pooled guide RNA (gRNAs) library [75,79,80]. The gRNAs direct the Cas9 enzyme for DNA double strand cleavage, while different kinds of mutations might be introduced by the error-prone Non-Homologous End Joining (NHEJ) DNA repair mechanism [81,82]. Therefore, it is possible to perturb many target genes simultaneously by using predesigned pooled gRNAs.

CRISPR-based gene activation (CRISPRa) and inhibition (CRISPRi), representing more powerful tools than RNA interference (RNAi) libraries, are highly useful in screening for gain and loss of function studies, respectively. Positive and negative selection using CRISPR libraries can clarify drug resistance mechanisms and survival-essential genes as drug targets [83]. CRISPR-Cas9 screening is currently employed to systematically investigate genes associated with lethal phenotypes and subsequently, propose drug targets at the gene and/or protein levels and accordingly identify likely repositionable drug candidates [84,85,86,87] (Figure 4). 

Fused genes are two independent genes formed through structural rearrangements, the transcription of adjacent genes, or the splicing stage of pre-messenger RNA, which may lead to the dysfunctionality of modified genes and deregulation of associated genes, and afterward, the probable overexpression of oncogenes and/or decreased expression of tumor suppressor genes [88].

One of the first large-scale systematic analyses of thousands of gene fusions in human cancers analyzed 1034 samples from 1011 unique cancer cell lines to primarily define gene fusions [9]. Subsequently, whole-genome CRISPR-Cas9 drop-out screening was applied to identify gene fusions required for cancer cell fitness [84,85,86,87,89]. Similar techniques may be applied to identify repositionable anti-cancer drugs and propose new treatments by determining several fused genes as potential targets.

## 5. Systems Biology Application in Drug Repositioning

Wet lab high-throughput approaches examine all possible targets to find a repurposable drug [90]. In this regard, computer-assisted systematic drug repurposing methods may overcome the limitations associated with wet lab high-throughput techniques using focused resources based on knowledge of the gene/protein target and literature [91].

Personalized drug repurposing may now be employed systematically [90] by employing a systems medicine approach to handle the massive amounts of medical information which is accumulated in DBs [92]. Five major categories may be employed to classify in silico drug repurposing methods, namely network-based, machine learning, chemoinformatic-, bioinformatic-, and signature-based approaches [90].

### 5.1. Network-Based Methods in Drug Repositioning

Network-based methods integrate various high-throughput data to address issues in drug repositioning and drug combination. Specifically, they rely on known biological relationships of drug–gene–metabolite–protein–disease to generate networks, of which DisGeNET [93], the Therapeutic Target Database (TTD) [94], BindingDB [95], STRING [96], STITCH [97], and TCSBN [47] employ network-based approaches. Key studies in the field have highlighted how network-based methods may be employed for drug repositioning and drug combination. For instance, by quantifying the network proximity of disease genes and drug targets [98] and also drug targets and disease proteins [99] in the human (protein–protein) interactome, many novel drug–disease relationships for over 900 FDA-approved drugs and clinically efficacious drug combinations for specific diseases were identified. It should be noted that network-based methods are not alternatives, but complementary, to experimental approaches, capable of systematically capturing and characterizing coordinated cellular responses that would otherwise be technically challenging experimentally, further focusing clinical-phase attention on precise objectives [100]. To accomplish these tasks, network-based methods make use of various data, including phenotypic, drug-related, and molecular data, to construct various network types, such as Drug–Drug Interaction Networks, Drug–Target Interaction Networks, Drug–Disease Interaction Networks, and Multilayer Networks. By providing different biological characterizations, network-based methods are thus excellent frameworks in drug repositioning [100]. The idea behind network-based cluster approaches which can reveal the drug–disease or drug–target associations is that biological entities (e.g., diseases, drugs, and proteins) in the same module (also known as subnetworks or groups) share similar features in a biological network. Network-based approaches apply concepts from graph theory, where genes, drugs, diseases, or metabolites share biologically-demonstrated or putative associations. A module can consist of drug–disease, drug–drug, or drug–target relationships and this information can be extracted using the topological structure of a network. Density-based spatial clustering of applications with noise (DBSCAN) [101], Ordering Points To Identify the Clustering Structure (OPTICS), and the Statistical Information Grid (STING) [102] are some examples of the algorithms employed in network analyses. Importantly, in many cases, pleiotropic biological functions may occur; for instance, the proteins that are related to different cellular functions and consequently contribute to pleiotropic phenotypes when mutated [103]. Employing a *k*-means-based network cluster algorithm, chemical–protein interactions were recently used to uncover repositionable drugs in small-cell lung cancer [104].

Network-based propagation methods may be divided into local (i.e., consider information from a subset of the network’s nodes) [105] and global (i.e., derive information from the entire network) methods [106]. These approaches apply prerequisite information from the source node(s) to other network nodes [90,107]. The efficiency of these methods were proven for discovering disease–target, disease–gene, and disease–drug connections [108], and have been shown to provide a higher efficacy compared to other methods [106]. In addition, a comprehensive procedure based on formulating constraints on a score function related to the smoothness of the disease–gene network may lead to the discovery of disease–gene and disease–protein relationships [109]. This method could successfully predict gene targets for type 2 diabetes, Alzheimer’s disease, and prostate cancer [90]. Additionally, methotrexate, gabapentin, cisplatin, donepezil, and risperidone could be repurposed for a second indication through a disease–gene–drug network propagation approach [110]. These observations indicate that drug repurposing conducted through network-based approaches presents an efficient and systematic approach for proposing repurposable drugs. 

Computational methods may also derive information from genome-scale metabolic networks through the utilization of genome-scale metabolic models (GEMs). These are models that depict gene–protein–reaction relationships for entire metabolic genes in an organism, and can be simulated to predict metabolic fluxes for various systems-level metabolic studies [111]. For instance, a prostate cancer (PRAD)-specific GEM was reconstructed to investigate prostate cancer metabolism by the integration of global gene expression profiling of cell lines treated with more than 1000 different drugs [112]. As a result, among sulfamethoxypyridazine, azlocillin, hydroflumethiazide, and ifenprodil, as repositionable drugs proposed through an in silico cell viability assay, ifenprodil could be validated using an in vitro cell assay. In another study [113], GEMs used for drug design were assessed. In this regard, a list of human metabolites with KEGG [114] identifiers and their structures were retrieved using an updated human GEM [115]. Drug structures were compiled from DRUGB [20] and drug-metabolite pairs were analyzed. Finally, the effects of lipoamide analogs on MCF7 (a breast cancer cell line) and airway smooth muscle (ASM) cells were proposed and subsequently successfully demonstrated experimentally.

### 5.2. The Application of Machine Learning in Drug Repositioning

Machine-learning techniques have recently been employed to propose new drug candidates [116,117], where supervised approaches predict potential associations between approved drugs and health disorders [118]. For instance, the application of classification algorithms, including logistic regression, random forest, and support vector machine algorithms, has been conducted to construct prediction models and predict previously unknown pharmacological effects of different compounds [119]. In this case, in order to define the associations between drugs and diseases, four classes of drug-drug similarity (chemical structure, side-effects, gene ontology, and targets) and three categories of disease-disease similarity (phenotypes, human phenotype ontology, and gene ontology) were used. Finally, random forest methods exhibited the highest performance and could predict novel indications for 20 existing drugs and 31 compounds, which were subsequently validated using clinical trial data [119].

A swift escalation in the volume of biomedical literature has made it impossible to manually discover all meaningful connections between biomedical concepts, especially drug repositioning, which requires the integration of various aspects of biomedical knowledge. Hence, the automated text mining of literature applies natural language processing to transform unstructured text from various sources (databases, websites, books, reviews, and articles) into normalized, structured, and quantifiable data and their connections [120]. Co-occurrence methods reveal the association between two non-co-occurrence terms through discovering a third linking term that appears directly with each of them [121]. Recently, a comprehensive study was conducted to evaluate different drug repurposing strategies for Parkinson’s disease by text mining the scientific literature through comparing various methods, including the extraction of biomedical entities and their relationships, construction of a knowledge graph for Parkinson’s disease, knowledge representation learning, and machine learning-based prediction [122]. As a result, unstructured biomedical literature data were effectively transformed to structured data that could be directly used by modern computational methods, such as machine learning [122]. Text-mining methods were recently applied, together with network analyses, to create disease-specific drug–protein connectivity maps [123]. These approaches extract disease–protein relationships from molecular interaction networks through network mining and investigate PubMed abstracts to predict novel identify drug–protein associations. In this regards, diltiazem and quinidine, as hypertension and arrhythmia drugs, respectively, can be repurposed for Alzheimer’s disease, which has been confirmed by clinical evidence.

The application of deep learning as a promising approach for capturing complex and highly non-linear and heterogenous network structures has been actively explored. For instance [124], 10 drug–disease, one drug–side-effect, one drug–target, and seven drug–drug networks were integrated by a network-based deep-learning approach for in silico drug repurposing. Eventually, in addition to detecting an approved drug–disease association from the ClinicalTrials.gov database, repositionable drugs and novel drug targets were proposed for Alzheimer’s disease (e.g., risperidone and aripiprazole) and Parkinson’s disease (e.g., methylphenidate and pergolide).

### 5.3. Chemoinformatic-Based Methods in Drug Repositioning

Chemoinformatic approaches are one of the primary methods employed for screening repositionable drugs and create chemical libraries submitted to an in silico screen. These approaches are the basic components of many datasets, including DRUGB [20]. Using a similarity-based approach that correlates the anatomical therapeutic chemical classes of drug indications and their similarity, related therapeutic indications are proposed based on chemically similar drugs [125].

Ligand-based and structure-based approaches can be embedded in chemoinformatic-based drug repurposing. The story behind these two methods is that analogous compounds are likely to have similar biological properties [126] and proteins with almost the same structures tend to have similar functions and bind identical compounds [127]. However, the drugs identified through these two methods may share the same pathway with the template drug and accordingly, for a specific disease, if the template drug has already been used as a treatment, the role of these techniques becomes limited [128]. In addition, the lack of specificity is a serious drawback in this strategy. For instance, thalidomide has two chiral forms (the same chemical composition, but having mirrored structures). One can treat morning sickness, while the other form can have teratogen effects [129]. In this regard, molecular docking has been introduced as the most commonly used tool for structure-based virtual screening, in order to repurpose available drugs [130,131]. In this regard, Pinzi et al. repurposed cannabigerol, which is a non-psychoactive cannabinoid, as a low micromolar inhibitor of the enoyl acyl carrier protein reductase enzyme using an integrated ligand-based and structure-based study.

### 5.4. Bioinformatics-Based Methods in Drug Repositioning

From sequence alignment to domain similarity identification tools, a disease-centric approach is known as one of the principles of bioinformatical drug repositioning that tends to find diseases with the same dysfunctional proteins, which is often not clearly identified in experimental assays [90].

Local and global structural similarities are two different concepts; one can only be considered for similar binding sites of non-evolutionary proteins with variable folding and functions, and the other one is related to the whole structure of evolutionary proteins with considerable homology, respectively [132]. Ehrt et al. [127] claimed that the algorithms developed so far for binding site similarity assessments are not convincing since adequate benchmarking studies have not been developed. Accordingly, they suggested using a combination of in silico and experimental methods in a workflow to enhance the accuracy.

### 5.5. Signature-Based Drug Repositioning

The idea behind signature-based drug repositioning is that if the gene expression signature of a particular drug is opposite to the gene expression signature of a disease, that drug may have a potential therapeutic effect on the disease [133]. Drug signatures can be retrieved by comparing the gene expression profile of a cell line before and after its exposure to a specific drug. However, it should be considered that gene signatures are often used to address the off-targets of drugs in network pharmacology and these methods may not explain the relation between gene-expression alterations [134]. Some studies have tried to overcome this drawback through network-based analysis [133]. However, in complex disorders such as cancer [135], Alzheimer’s disease [136], and inflammatory bowel disease, promising results were achieved through signature-based methods [137]. The Connectivity Map (cMap) [138] project was later joined with the LINCS L1000 project [30], which are two comprehensive and pioneer databases in the field of signature-based drug repurposing providing the gene expression profiles of dozens of cell lines treated with thousands of small molecules. It is obvious that dealing with such a large dataset would be challenging for computational systems biologists who aim to analyze and visualize Big Data. The cMap project presents a valuable resource for conducting large-scale assays of many small molecules. However, it is not without limitations. First, while comprehensive, it has tested few cell lines. Second, it presents limited drug perturbation data and does not provide a comprehensive view of drug dosages and temporal dynamics upon drug exposure. Third, because of this, its application in other cell lines and biological samples is very limited and has been demonstrated to not be very robust [139]. However, extensively examining a high number of small molecules, with different dosages and time points, is an extremely expensive and time-consuming process. Seeking to overcome these issues, others sought to devise systematic approaches to test a large number of samples and compounds, while aiming to also provide a systematic view of intracellular behavior (L1000). However, doing so is not without its perils, as previously mentioned with regards to the computationally heavy process [139]. Accordingly, a systematic compound signature discovery pipeline, called Enrichment of Gene Effect to a Molecule (EGEM), was developed by Liu et al. [140], which was derived from a rank-based gene set enrichment analysis and showed a better performance in comparison to the original. To date, several drug candidates have been proposed thanks to signature-based studies of cMap and L1000 data, including KM-00927 and BRD-K75081836 as novel histone deacetylase inhibitors and mitomycin C as a topoisomerase IIB inhibitor [141].

## 6. Conclusions and Future Direction

The integration of large assays is crucial for accurate predictions of potential therapeutic targets from high-throughput assays. DBs store both computationally- and laboratory-based created data and, subsequently, systems medicine makes use of these resources to decipher potential disease mechanisms, targetable genes and pathways, and prognostic or theranostic features of biomarkers (Figure 5). Omic data have been coupled with essentiality screens, including RNAi screening or the more accurate CRISPR-Cas9 screens.

Several criteria are critical to facilitating DB usage in pharmaceutical science. First, one of the most important features of a database is that it has to be publicly available. Second, the availability of data after publication is another important criterion. Third, being updated in both data and interface aspects is crucial for drug discovery. Fourth, having an application programming interface (API) can help expert users to conveniently retrieve their required information, and many currently devise APIs to enable computational scientists to easily retrieve and analyze their data (e.g., LINCS L1000 and CheMBL). Fifth, data and metadata should be uniformly deposited and easily retrievable. Sixth, fully downloadable data are necessary for several experts to develop their own methods. LINCS L1000, ENCODE, TTD, and DSDB are instances of databases with data that are available to download. Seventh, internal tools such as the drug repositioning package of LINCS L1000 and SCYP can facilitate the use of databases. Eighth, advanced searches and queries increase the capabilities of databases, for instance, DrugDB and LINCS L1000 (Slicer, LINCS Canvas Browser, and L1000 Viewer). Researchers can make use of the feature list and characterization provided here (Table 1 and Appendix A, and Figure 2) as a starting point to identify suitable datasets and other information for their research purposes.

Previously FDA-approved chemicals as drugs may be repurposed for new targets and their mode of action and side effects can be investigated. Systematic drug repurposing requires a comprehensive collection of all these small-molecule drugs. There are many comprehensive databases encompassing various data types, and integrated perspectives such as those given by the DrugBank may facilitate drug repositioning. Systems biology and systems medicine may assist in this effort by applying machine learning and integrative approaches to predict repurposable drugs based on data from different sources. In this case, method and algorithm development seems necessary to escalate the efficiency of in silico screenings.

## Figures and Tables

**Figure 1 cancers-12-02694-f001:**
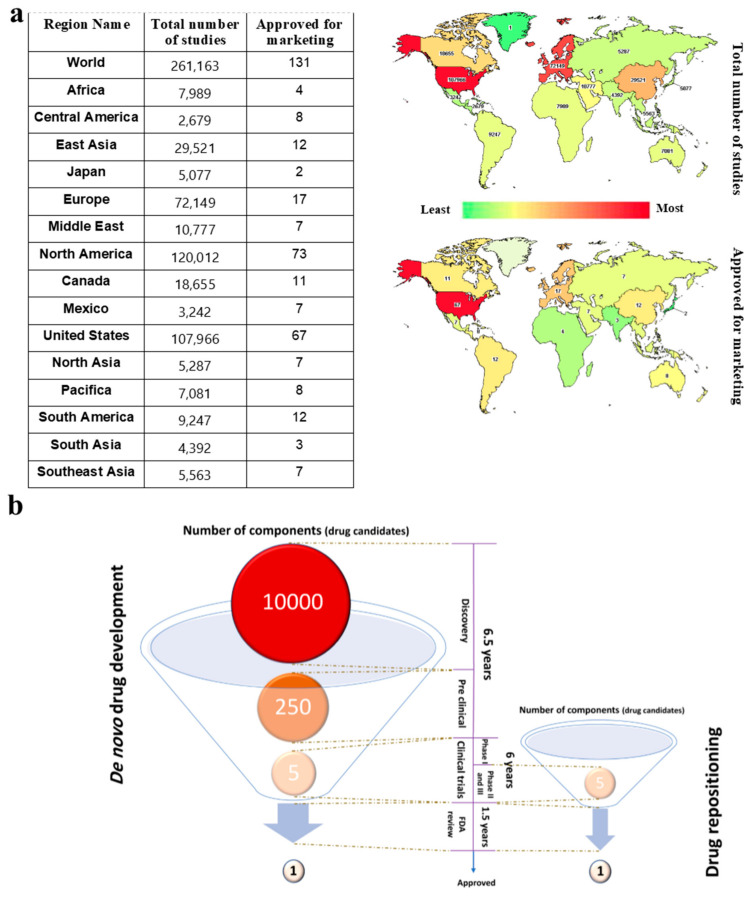
(**a**) Drug repositioning significantly cuts drug development costs and lengths of time. The distribution of the total number of conventional drug development studies and approved drugs for marketing is shown in the table and map. The USA, north North America, and Europe have the highest number of ongoing clinical trials [5]. (**b**) Two different methods for drug development are compared. The number of components (biological and/or chemical drug candidates), as input into de novo drug development, is 2000 times higher than drug repositioning, while drug repositioning may start in more advanced clinical trial phases and save several years when trying to find new treatments for emerging diseases.

**Figure 2 cancers-12-02694-f002:**
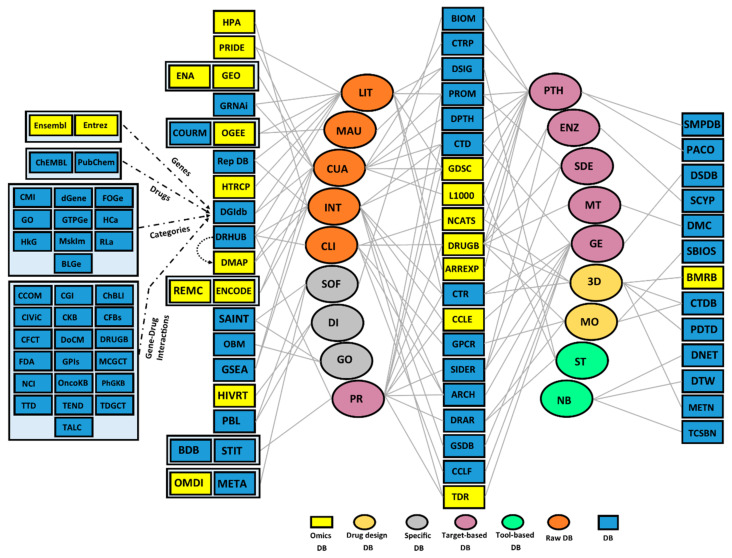
Key databases as primary resources for drug repositioning. Databases (DBs) can be divided into five classes, including Drug Design DB, Specific DB, Target-based DB, Tool-based DB, and Raw DB. The content of each DB is shown with an edge to the subcategories. Refer to Appendix A for a full description. DBs with omics and high-throughput datasets are colored in yellow. Subcategories: 3D: three-dimensional information; CLI: clinical; CUA: curation; DI: disease; ENZ: enzyme; GE: gene target; GO: geographic; INT: integration; Lit: literature; MAU: manual; MO: molecular information; MT: multidrug; NB: net-base; PR: protein target; PTH: pathway; SDE: side effect; ST: simple tool; TR: software. Databases: ACHI: Achilles; ARREXP: ArrayExpress; BLGe: BaderLabGenes; BIOM: Biomodels; BMRB: Biological Magnetic Resonance Data Bank; CCOM: Cancer Common; CCLF: cancer cell line factory; CCLE: Cancer Cell Line Encyclopedia; CFBs: Clearity Foundation Biomarkers; CFCT: Clearity Foundation Clinical Trial; CGI: Cancer Genome Interpreter; ChBLI: The ChEMBL Bioactivity Database; CIViC: Clinical Interpretations of Variants in Cancer; CKB: The Jackson Laboratory Clinical Knowledgebase; CTDB: Cheminformatic Tools and Databases; CMI: Caris Molecular Intelligence; CORUM: comprehensive resource of mammalian protein complexes; CTR: Cancer Therapeutics Response Portal; CTD: The Comparative Toxicogenomics Database; CTRP: Clinical Trials Reporting Program; dGene: The Druggable Gene List; DGIdb: drug–gene interaction database; DMAP: Dependency Map; DMC: Drug Map Central; DNET: Drug-Disease Network database; DoCM: Database of Curated Mutations; DPTH: Drug Pathway database; DRUGB: DrugBank; DRHUB: drug repurposing hub; DRAR: Drug Repurposing Adverse Reaction; DSDB: Drug Signatures Database; DSIG: DrugSig; DTW: Drug Target Web; ENCODE: Encyclopedia of DNA Elements, Entrez, Ensembl; FDA: FDA Pharmacogenomic Biomarkers; FOGe: Foundation One Gene; GEO: Gene Expression Omnibus; GPIs: Guide To Pharmacology Interactions; GPCR: G-protein-coupled receptors assay bank; GDSC: Genomics of Drug Sensitivity in Cancer; GO: The Gene Ontology; GRNAi: GenomeRNAi; GSEA(MSigDB): Gene Set Enrichment Analysis; GSDB: Gene Set Database; GTPGe: Guide To Pharmacology Genes; HCa: Hingorani Casas; HIVRT: HIV Drug Resistance Database; HkG: Hopkins Groom; HTRCP: host transcriptional response connectivity map; L1000: Library of Integrated Network-based Cellular Signatures (LINCS) L1000; MCGCT: My Cancer Genome Clinical Trial; METN: Metlin; META: Metabolic Atlas; MskIm: Memorial Sloan Kettering IMPACT; NCI: Cancer Gene Index; NCATS: National Center for Advancing Translational Sciences; NNFIN: network-based similarity finder; ODB: Ontario database; OGEE: Online GEne Essentiality; OMDI: Omics Discovery Index (omicsdi); OncoKB: A Precision Oncology Knowledge Base; PACO: pathway commons; HPA: Protein Atlas; PBL: ProBiS-ligands; PDTD: Potential Drug-Target Database; PhGKB: The Pharmacogenomics Knowledgebase; PRIDE: PRoteomicsIDEntifications (PRIDE) database; PROM: Promiscuous; RepDB: Repurpose DB; RLa: RussLampel; SAINT: Significance Analysis of INTeractome; SBIOS: Swiss BIOisostere; SCYP: Super Cytochrome P450; SIDER: Side Effect Resource; SMPDB: Small Molecule Pathway Database; TALC: Targeted Agents in Lung Cancer; TCSBN: tissue- and cancer-specific biological networks; TDR: Tropical Diseases Research; TDGCT: The Druggable Genome (TDG) Clinical Trial; TEND: trends in the exploitation of novel drug targets; TTD: Therapeutic Target Database.

**Figure 3 cancers-12-02694-f003:**
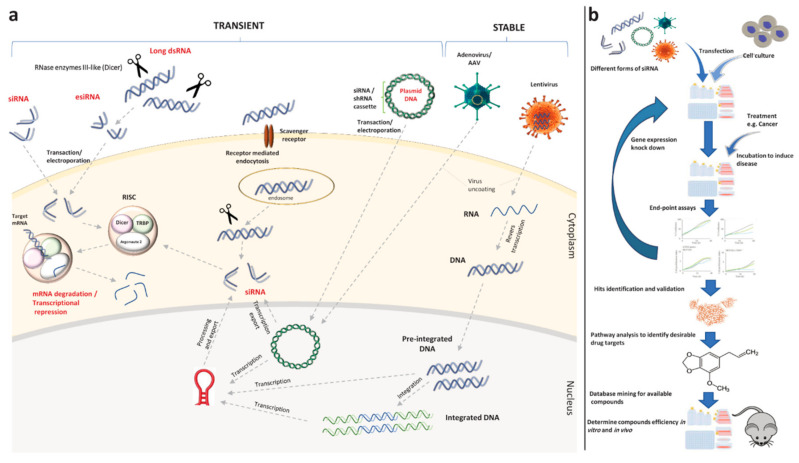
Various RNA interference (RNAi) constructs used for temporary or permanent gene silencing and a general workflow of small interfering RNA (siRNA) screening towards drug repositioning (adapted from [72]). (**a**) siRNA, endoribonuclease-prepared siRNA (esiRNA), and long double-strand RNA (dsRNA), as naked RNAs, can be introduced to the cell through many pathways, such as lipid-based transfection reagents or electroporation, or active uptake by target cells through receptor-mediated endocytosis. Inside the cytoplasm, dsRNA is cleaved by endonucleases. Then, the obtained siRNA, along with the directly introduced siRNAs and siRNAs coming from the nucleus of hard-to-transfect cells which were achieved from viral vectors carrying siRNA or short hairpin RNA (shRNA) expression cassettes, are loaded into the RNA-induced Silencing Complex (RISC) to facilitate gene expression knock down. (**b**) Drug targets and available compounds are determined by in silico pathway analyses and database mining after two-step in vitro screening. Prior to clinical analysis, the efficiency of these drug candidates can be validated by in vitro and in vivo approaches.

**Figure 4 cancers-12-02694-f004:**
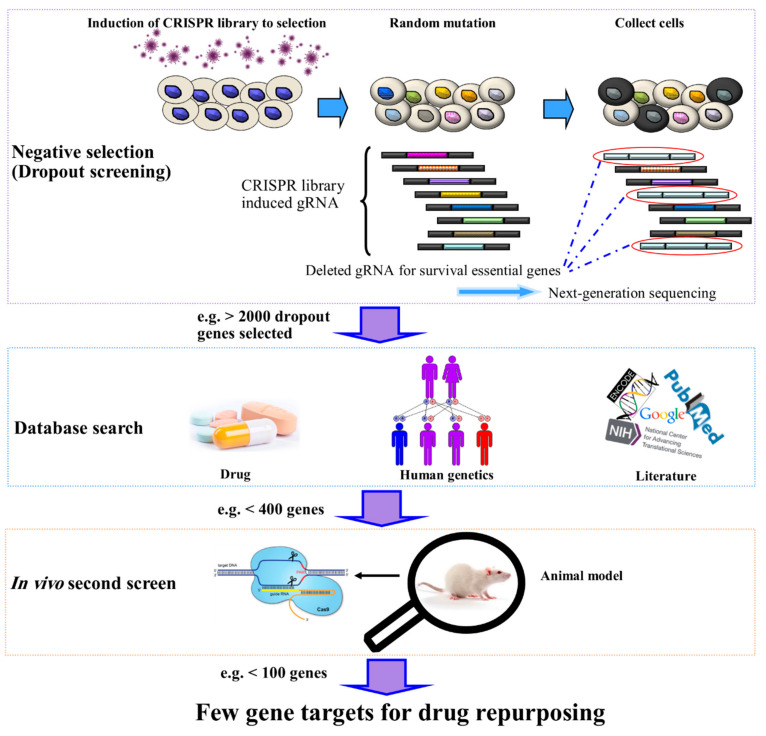
Pipeline of drug repositioning for a specific cancer or other disease. First, clustered regularly interspaced short palindromic repeats (CRISPR)-Cas9 dropout screening is applied for a negative selection of survival essential genes. Second, the number of dropped out selected genes is diminished by collecting information from databases and literature reviews. Third, in vivo CRISPR-Cas9 screening of an animal model of cancer or another disease will result in a few gene targets. Finally, FDA-approved drugs or those which have passed primitive phases of clinical trials could be tested as inhibitors for discovering essential survival genes.

**Figure 5 cancers-12-02694-f005:**
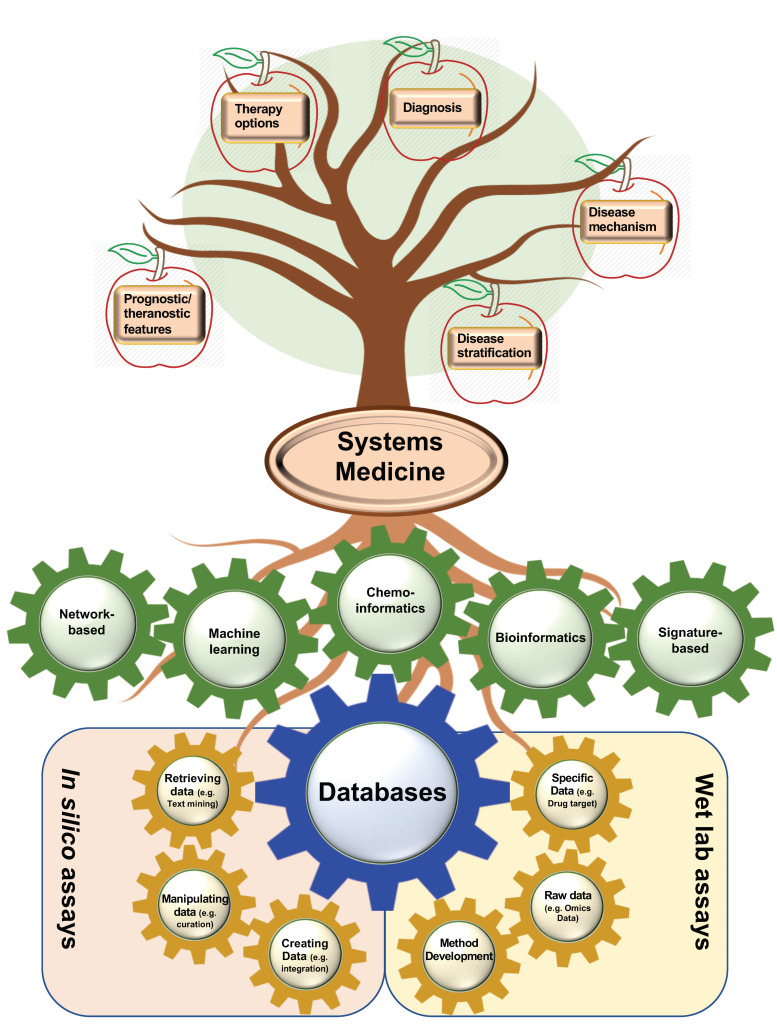
From bottom to the top, this is the whole story of current efforts to find repositionable drug candidates for emerging disease. DBs, as the foundation of drug repositioning, can be created from various computational and experimental approaches. The second way of categorizing DBs (assay-based classification) is depicted here as three in silico and three laboratory-based approaches in yellow gears to construct a DB. Afterwards, DBs can pave the way for various categorizations of systems medicine (shown in green gears) to decipher many probabilities in the case of disease mechanisms; disease stratification; therapy options; and diagnostic, prognostic, or theranostic features of biomarkers.

**Table 1 cancers-12-02694-t001:** DBs and their associated omic data as useful resources for drug repositioning.

Database	Type of Omics data	Link	Ref.
ARREXP	Functional genomics data	https://www.ebi.ac.uk/arrayexpress/	[17]
BMRB	Proteomics, metabolomics	http://www.bmrb.wisc.edu/	[18]
CCLE	Genomic data	https://portals.broadinstitute.org/ccle	[19]
DRUGB	Transcriptomics	https://www.drugbank.ca/	[20]
DMAP	Transcriptomics, proteomics	https://depmap.org/portal/	[21]
ENA	Genomics, transcriptomics	https://www.ebi.ac.uk/ena	[22]
ENCODE	Genomics, transcriptomics	https://www.encodeproject.org/	[23]
ENTREZ	Genomics, transcriptomics, proteomics, taxonomics	https://www.ncbi.nlm.nih.gov/Class/MLACourse/Original8Hour/Entrez/	[24]
ENSEMBL	Genomics, transcriptomics	https://www.ensembl.org/index.html	[25]
GDSC	Transcriptomics	https://www.cancerrxgene.org/	[26]
GEO	Functional genomics, transcriptomics, proteomics, phenomics	https://www.ncbi.nlm.nih.gov/geo/	[27]
HIVRT	Genomics, phenomics	https://hivdb.stanford.edu/	[28]
HTRCP	Transcriptomics	http://biotech.bmi.ac.cn/papers/2015/luhan.html	[29]
L1000	Transcriptomics	http://www.lincsproject.org/LINCS/tools/workflows/find-the-best-place-to-obtain-the-lincs-l1000-data	[30]
META	Metabolomics	https://metabolicatlas.org/	[31]
OMDI	Genomics, proteomics, transcriptomics and metabolomics	https://www.omicsdi.org/	[32]
HPA	Proteomics	https://www.proteinatlas.org/	[33]
PRIDE	Proteomics	https://www.ebi.ac.uk/pride/archive/	[34]
NCATS	Chemical genomics	https://ncats.nih.gov/	[35]
OGEE	Genomics, transcriptomics	http://ogee.medgenius.info/browse/	[36]
REMC	Epigenomics	http://www.roadmapepigenomics.org/	[37]
TDR	Genomics	https://tdrtargets.org/	[38]

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
