# Peer review of "Applications of Genome-Wide Screening and Systems Biology Approaches in Drug Repositioning"

_cancers, 2020, doi:10.3390/cancers12092694_

Round 1

Reviewer 1 Report

The authors present a very interesting review to drug repurposing. The manuscript is very well written and organized and I believe it to be very useful for those working in the field. I think that the bibliography section on network based methods should also contain a a recent general review such as: 

Conte, F., Fiscon, G., Licursi, V., Bizzarri, D., D'Antò, T., Farina, L., Paci, P. A paradigm shift in medicine: A comprehensive review of network-based approaches (2020) Biochimica et Biophysica Acta - Gene Regulatory Mechanisms, 1863 (6), art. no. 194416

Author Response

Reviewer 1:

The authors present a very interesting review to drug repurposing. The manuscript is very well written and organized and I believe it to be very useful for those working in the field.

    • I think that the bibliography section on network based methods should also contain a a recent general review such as: 

Conte, F., Fiscon, G., Licursi, V., Bizzarri, D., D'Antò, T., Farina, L., Paci, P. A paradigm shift in medicine: A comprehensive review of network-based approaches (2020) Biochimica et Biophysica Acta - Gene Regulatory Mechanisms, 1863 (6), art. no. 194416

We are very thankful for the reviewer’s positive comment. We have added a brief section of the valuable article mentioned as follow:

Page 16, lines 351-360

“ It should be noted that network-based methods are not alternatives but complementary to experimental approaches, capable of systematically capturing and characterizing coordinated cellular responses that would otherwise be technically challenging experimentally, further focusing clinical-phase attention on precise objectives[88].To accomplish these tasks, network-based methods make use of various data including phenotypic, drug-related, and molecular data, to construct various network types such as  Drug-Drug interaction Networks, Drug-Target interaction Networks, Drug-Disease Interaction networks, and Multilayer Networks. By providing different biological characterizations, network-based methods are thus excellent frameworks in drug repositioning[88].”

Reviewer 2 Report

The authors provide a strong overview  of database-driven screening of biological, chemical, and pharmacology databases that could be used for drug discovery. The point that could have been made more strongly is the utility of the drug repurposing.

A point that the authors should stress is rationale v. empiric drug repositioning. A strong enough case is not made for empiric drug (re)discovery.

The list of internet available databases is invaluable -- but there should be some editorializing of which ones are really good

The sections decribing the biology and the database mining would be greatly and invaluably strengthened with examples of drug repurposing. In particular, discussion of the  utility (and the disappointing lack of robust discoveries) from the Connectivity Map should be expanded. 

minor point line 45, 48 novel drugs [were approved in 2019] out of 318,901 clinical trials is misleading,. how many novel drugs are in those clinical trials. 

minor point line 52, capitalize N in North America. In addition, there are no data on "investment" in the Figure 1a.

minor point on line 60, it may be confusing "malaria with low funding and demand". This reviewer thinks that there is a high demand for malaria therapy.  This is a major focus by the Gates Foundation and its biology has garnered publication in Nature over the past several years. Yes, there is low funding from non-Gates Fdn sources. 

Author Response

  • Reviewer 2:
  • The authors provide a strong overview of database-driven screening of biological, chemical, and pharmacology databases that could be used for drug discovery. The point that could have been made more strongly is the utility of the drug repurposing.
    • A point that the authors should stress is rationale v. empiric drug repositioning. A strongenough case is not made for empiric drug (re)discovery.

We are very grateful to the reviewer’s constructive comments, which greatly assisted us in improving our manuscript. We have revised the manuscript accordingly as follows:

Page 3, lines 42-60

“Empirical investigation of repositionable drugs leads to rescue the utility of many FDA approved chemicals [2,3]. Empirical drug repositioning represents the process of finding an unknown target for a known drug in vitro. It has been proposed that empirical drug repositioning could be as a starting point for the process of drug repurposing by in vitro screening of known drugs or drug-like molecules to identify and validate candidates for repositioning which brings these advantages: the knowledge for a potential new disease setting will be increased; serendipitous, or hypothesis-free, assessment of compounds will be achieved by testing multiple compounds with different modes of action; and it brings data-driven choices for further investigations in more complex phenotypic or in vivo tests. As an example, an FDA-approved drug library of 640 compounds was screened and 10 of them, including tamoxifen and raloxifene, proposed as compounds that are able to protect hair cell loss in response to the known inner ear mechanosensory hair cell toxins such as neomycin, gentamicin, kanamycin, and cisplatin which cause hearing impairment and balance disorders [2]. In rational drug repurposing, the target is known from the beginning and subsequently, the goal would be to find a previously FDA approved compound which would interact with the target of interest [4]. While in empirical drug repositioning a large numbers of combinatorial compounds are prepared and tested, rational drug repurposing reduce the number of candidates in the early stages of investigation and identify entities with a high probability of success [5].”

  • The list of internet available databases is invaluable -- but there should be some editorializing of which ones are really good

While we understand the reviewer’s comment, it is challenging to classify what are good or bad databases, as this would depend on one’s research purpose. For this reason, we aimed to provide more objective information of each database, including data availability, number and type of data and assays, and also distinguished omics-based and non-omics databases to leave the decision to the reader on which database is best suited for their research needs. For instance, some databases are more comprehensive and have various types of data (ENCODE) while others have fewer data types (LINCS L1000), and yet others are based on data curation (DGidb). However, one would arguably make use of any of the three.

We have revised the following paragraph in the conclusion and future direction section to stress the reviewer’s point (Page 22-23, lines 517-531):

Several criteria are critical to facilitate DB usage in pharmaceutical science. First, one of the most important features for a database is it to be publicly available. Second, availability of data after publication is another important criterion. Third, being updated in both data and interface aspects is crucial for drug discovery. Fourth, having an application programming interface (API) can help expert users to conveniently retrieve their required information, and many currently devise APIs to enable computational scientists to easily retrieve and analyse their data (e.g. LINCS L1000 and CheMBL). Fifth, data and metadata should be uniformly deposited and easily retrievable. Sixth, fully downloadable data is necessary for several experts to develop their own methods. LINCS L1000, ENCODE, TTD, DSDB are instances of databases with available data for download. Seventh, internal tools like drug repositioning package of LINCS L1000 and SCYP can facilitate the use of databases. Eighth, advanced search and query increase the capabilities of databases, for instance DrugDB and LINCS L1000 (Slicer, LINCS Canvas Browser, and L1000 Viewer). Researchers can make use of the feature list and characterization provided here (Table 1, Table S1, and Figure 2) as a starting point to identify suitable datasets and other information for their research purposes.”

  • The sections decribing the biology and the database mining would be greatly and invaluably strengthened with examples of drug repurposing. In particular, discussion of the utility (and the disappointing lack of robust discoveries) from the Connectivity Map should be expanded. 

We have added a more detailed description about connectivity map in the “signature-based drug repositioning” section. This revision follows (Page 21-22, lines 490-502):

“It is obvious that dealing with such a large dataset would be challenging for computational systems biologists who aim to analyze and visualize Big Data. The cMap project presents as a valuable resource for large-scale assay of many small molecules. However, it is not without limitations. First, while comprehensive, it has tested few cell lines. Second, it presented limited drug perturbation data and does not provide a comprehensive view of drug dosages, temporal dynamics upon drug exposure. Third, because of this, its application in other cell lines and biological samples is very limited and has been demonstrated to not be very robust [128]. However, extensively examining a high number of small molecules, with different dosages and time points is an extremely expensive and time-consuming process. Seeking to overcome these issues, others sought to devise systematic approaches to test a large number of samples and compounds, while aiming to also provide a systematic view of intracellular behavior (L1000). However, doing so is not without its perils as we mentioned about computationally heavy process [128]”

(Page 18-19, lines 421-428):

“Recently, a comprehensive study was done to evaluate different drug repurposing strategies for Parkinson’s disease by text mining the scientific literature through comparing various methods including the extraction of biomedical entities and their relationships, construction of a knowledge graph for Parkinson’s disease, knowledge representation learning and machine learning-based prediction [110]. As a result, unstructured biomedical literature data were effectively transformed to structured data that could be directly used by modern computational methods such as machine learning [110].”

  • minor point line 45, 48 novel drugs [were approved in 2019] out of 318,901 clinical trials is misleading,. how many novel drugs are in those clinical trials. 

Unfortunately, we do not have information between repurposed and novel drug identification in clinical trials. We thus revised the following sentence to reflect this (Page 4, lines 66-68):

“In the beginning of 2020 there were 261,163 ongoing clinical trial studies (February 2020, www.clinicaltrials.gov) distributed all over the world (Figure1.a). In 2019, there were 318,901 ongoing clinical trials and while it is unclear how many of these clinical trials featured de novo drug design, only 48 novel drugs were approved for usage by FDA (www.fda.gov).”

  • minor point line 52, capitalize N in North America. In addition, there are no data on "investment" in the Figure 1a.

We have revised the figure caption accordingly (Pages 32, lines 926-927)

  • minor point on line 60, it may be confusing "malaria with low funding and demand". This reviewer thinks that there is a high demand for malaria therapy.  This is a major focus by the Gates Foundation and its biology has garnered publication in Nature over the past several years. Yes, there is low funding from non-Gates Fdn sources. 

We have corrected the context accordingly (Pages 4, lines 73).

Round 2

Reviewer 1 Report

accept as is